# Prevalence and risk factors of scabies among orphans: A cross-sectional study in Bangladesh

Md Abdur Rafi[1], Md. Golam Hossain[2], Md. Rashidul Hasan[3],
Mohammad Jahid Hasan [iD][1,4]*

1 Tropical Disease and Health Research Center, Dhaka, Bangladesh, 2 Health Research Group, Department of Statistics, University of Rajshahi, Rajshahi, Bangladesh, 3 US-Bangla Medical College Hospital, Dhaka, Bangladesh, 4 Pi Research and Development Center, Dhaka, Bangladesh

* dr.jahid61@gmail.com

## Abstract

### Introduction

Scabies is one of the most common neglected tropical diseases predominantly affecting children from low socio-economic background. This study was conducted to determine the prevalence and associated factors of scabies among the children living in the orphanages of Bangladesh.

### Methods

This was a cross-sectional study among children in selected orphanages in Dhaka city of Bangladesh from May to December 2023. The participants were screened for scabies according to the diagnosis criteria of the International Alliance for the Control of Scabies guideline (IACS, 2020). A logistic regression model was used to determine the factors associated with scabies among the children.

### Results

A total of 471 children living in five selected orphanages of Dhaka city of Bangladesh were screened for scabies. Majority of the children living in orphanages were female with a mean age of 11.9 (SD 3.24) years. The overall prevalence of scabies among the children was 31.6% (mild 60.4%, moderate 28.2% and severe 11.4%). Male gender (aOR 2.76, 95% CI 1.14, 7.45, p-value 0.032), increased number of children per room (aOR 1.07, 95% CI 1.03, 1.12, p-value 0.001), and history of pruritus in the close contacts or peers (aOR 1.89, 95% CI 1.04, 3.46, p-value 0.038) were associated with a higher odds of being infected by scabies.

### Conclusion

One-third of the children living in the orphanages were suffering from scabies. Male children and those who live in crowded facilities and had close contact with infected peers were at higher risk of scabies.

**Data availability statement:** All relevant data are within the manuscript and its Supporting information files.

**Funding:** Research reported in this publication was supported by the Bangladesh Medical Research Council (BMRC) under Award Number BMRC/Research Grant Revenue/2022-2023/34(1-19) to MGH. The content is solely the responsibility of the authors and does not necessarily represent the official views of the BMRC. The funder had no role in study design, data collection and analysis, decision to publish, or preparation of the manuscript.

**Competing interests:** The authors have declared that no competing interests exist.

## Author summary

Scabies is a common but often overlooked skin disease that affects children, particularly those from disadvantaged backgrounds. This study investigated how widespread scabies is among children living in orphanages in Dhaka, Bangladesh, and identified the factors that contribute to its spread. We screened 471 children across five orphanages and found that nearly one-third (31.6%) were affected. Boys, children living in crowded rooms, and those who had close contact with others experiencing itching were more likely to have scabies. By addressing overcrowding and improving awareness about scabies transmission, policymakers and healthcare providers can help protect vulnerable children from this highly contagious disease.

## Introduction

Scabies, a highly contagious skin disease mainly affecting children, is caused by a parasitic mite *Sarcoptes scabiei* [1]. This disease is transmitted through direct and prolonged contact with infected skin or by using contaminated personal objects [2]. The predominant clinical manifestations of scabies include debilitating itching and scratching, often lasting for months in untreated patients [3]. Complications among these patients are frequent due to secondary bacterial infection, ranging from impetigo, abscesses, and cellulitis [4]. Scabies also causes significant morbidity, resulting from sleep disruption, concentration difficulties, and impaired productivity [5].

Being one of the most common Neglected Tropical Diseases (NTDs) with skin manifestations, scabies contributes to a substantial public health burden globally, resulting from high case numbers, disease complications, and containment costs [6,7]. Over 175 million people worldwide are infected by this disease, mostly in lower- and middle-income countries of Sub-Saharan Africa and South-East Asia, contributing to almost 4.84 million disability-adjusted life-years (DALYs) annually [8,9].

Scabies disproportionately affects children in disadvantaged settings. Overcrowded living conditions, malnutrition, shared sleeping arrangements and sharing of clothes and towels with those who are already infected are common risk factors for this disease [10,11]. Children in lower socioeconomic settings, especially those living in unhygienic and crowded areas like urban slums and orphanages, are most vulnerable to this disease [11–13]. In these children, infestations often spread rapidly due to close contact and overcrowding. However, treating this group is challenging due to delayed diagnosis, inadequate treatment, malnutrition, associated allergic and bacterial infections, and insufficient follow-up [14].

Communal settings and closed residential environments, like orphanages, where a large number of children live together in the same building or compound, are highly vulnerable for scabies due to factors like crowded living conditions and close contacts with infected peers [15]. In Bangladesh, a lower-middle-income country of south-east Asian Region, there are more than four hundred thousand orphan children

[16]. Among them, approximately 9500 orphans are accommodated in 73 government-run orphanages while the private orphanages accommodate far larger number [17]. Majority of these private orphanages depends on the community donation which is often collected by the children from the households. A lot of these facilities are linked to religious schools where other children are also enrolled. Moreover, the residents of orphanages often attend community social and religious customs like funerals, prayers etc. These intense social interactions of the children living in orphanages could make them a potential source of scabies for the community. Hence, prevention and treatment of scabies in orphanages might help to reduce community burden of scabies as well. Understanding the epidemiology of scabies in orphanages would allow design effective prevention strategies. Therefore, this study aimed to determine the prevalence and associated factors of scabies among children in orphanages in Dhaka city, Bangladesh.

## Methods

### Ethics approval and consent to participate

The study protocol was reviewed and approved by Institutional review board of Public Health Foundation, Bangladesh (PHFBD-ERC-SG04/2023). The all authors declare no human subjects were harmed and the procedures followed were in accordance with the ethical standards and regulations established by the Helsinki Declaration of the World Medical Association. Formal informed written consent was obtained from the parent/guardian of the minor participants involved in the study.

### Study design and setting

This was a cross-sectional study conducted in orphanages of Dhaka city of Bangladesh from July to December 2023. The orphanages were selected conveniently based on the inclusion criteria of accommodating at least 50 children of both male and female gender during the study period.

### Participants

All the children aged between 3 and 18 years residing in the selected orphanages were considered as the study population. Inclusion criteria for the participants were: both male and female children aged between 3 and 18 years residing in the orphanages for at least six months.

### Sample size determination and sampling

Sample size for this study was calculated using the single population proportion formula: $n = z^2 \, p(1-p)/d^2$, where $z = 1.96$ for 95% confidence level, $p$ = estimated prevalence of scabies and $d$ = margin of error. For this study, the following assumptions were considered: estimated prevalence of scabies 18% reported by a previous study conducted in boarding schools of Cameroon [18], margin of error at 5%, non-response rate as 10%. The formula provided that a sample size of 252 would be enough for this study. However, we approached a total of 496 children according to our inclusion criteria by consecutive sampling from the selected orphanages, among whom a number of 471 children were included in the final analysis.

### Data collection

An official written permission was obtained from the authority of the orphanages after explaining the objectives and procedure of the study. As the study population was from minor age group (age < 18 years) and the orphanage authority was the legal guardian, this permission was considered as the assent for including the children.

Afterwards, all participants were interviewed by trained research assistants and examined by a registered physician. Suspected cases for scabies those were not confirmed by the physician, were reviewed by a specialist dermatologist.

A semi-structured case record form comprising of sociodemographic information, personal practice related information and clinical information was used for data collection (S1 File). Examination of the skin primarily focused on most commonly affected regions by scabies. However, examination of sensitive areas such as the groin, buttocks, breasts, and torso were not routinely done for all the children due to the challenges of assessing private body parts in a field setting, particularly among female children, most of whom adhere to Islamic practices and customarily wear veils. Evidence indicates that a limited examination focusing on common sites, such as the hands, feet, and lower legs, can detect approximately 90% of scabies cases [19]. Participants presenting with clinical manifestations of scabies in these regions along with complaints of itching in their private parts were asked whether they had similar lesions in those sensitive areas. If they confirmed the presence of lesions, a physical examination was conducted in a private setting, ensuring proper privacy, and in the presence of a female physician for female children and male children under the age of 12 years, and a male physician for other children. All the diagnosed cases of scabies received standard management for free of cost from the research team.

## Diagnosis of scabies

Scabies was diagnosed in this study based on criteria for scabies diagnosis developed by the IACS, 2020 consensus criteria [20].

| A | Confirmed scabies meets at least one of the following criteria: |
|---|---|
| A1 | Mites, eggs or feces on light microscopy of skin samples |
| A2 | Mites, eggs or feces visualized on an individual using a high-powered imaging device |
| A3 | Mite visualized on an individual using dermoscopy |
| **B** | **Clinical scabies meets at least one of the following criteria:** |
| B1 | Scabies burrows |
| B2 | Typical lesions affecting male genitalia |
| B3 | Typical lesions in a typical distribution and two history features |
| **C** | **Suspected scabies meets one of the following criteria:** |
| C1 | Typical lesions in a typical distribution and one history feature |
| C2 | Atypical lesions or atypical distribution and two history features |

In this study, we used the subcategory of B (clinical scabies) and C (suspected scabies) for diagnosing scabies.

The severity of scabies was defined based on the number of lesions counted as mild (1–10 lesions), moderate (11–49 lesions) and severe scabies (>50 lesions) [21].

## Statistical analysis

Following data collection using a paper-based case record form (CRF), all data were anonymously entered into Microsoft Excel 365 for further analysis. The prevalence of scabies was determined among the participants according to the 2020 IACS criteria as mild, moderate and severe scabies [20]. A multiple logistic regression analysis was conducted to identify factors associated with scabies in orphaned children, with odds ratios (ORs) and 95% confidence intervals (CIs) reported. Prior to model interpretation, multicollinearity was assessed using the variance inflation factor (VIF). While no variable exhibited high collinearity (VIF > 5), some variables, such as number of children in the orphanage, use of the same toilet, and sleeping place, showed moderate VIF values (VIF > 4), suggesting potential collinearity concerns. However, these variables were retained in the model considering their theoretical significance. Besides, we initially included site (orphanage) as a factor to adjust for potential differences between study locations. However, site did not significantly influence the

model estimates, and its effect was negligible. Therefore, it was excluded from the final model. STATA version 17.0 was used for statistical analysis.

## Results

### Baseline characteristics

A total of 471 orphaned children from five orphanages of Dhaka city of Bangladesh were included in this study. Their mean age was 11.9 (SD 3.24) years and majority of them were female (almost 76.9%). An orphanage accommodated a median number of 94 children. The median number of children sharing a room and a toilet were 10 and 15 respectively. Almost 16% of the children reported that they shared bed for sleeping with other children while 26% reported that they shared their personal items with others (Table 1).

**Table 1. Baseline characteristics of the children living in orphanages (n = 471).**

| Characteristic | n (%) or mean (SD) or median (IQR) |
|---|---|
| Age (years), mean (SD) | 11.89 (3.24) |
| Gender, n (%) | |
| Female | 362 (76.8) |
| Male | 109 (23.1) |
| Number of orphans in the orphanage, median (IQR) | 94 (85.0) |
| Number of children per room, median (IQR) | 10 (17.0) |
| Number of children sharing same toilet, median (IQR) | 15 (11.0) |
| Sleeping place, n (%) | |
| On bed | 201 (42.7) |
| On floor | 270 (57.3) |
| Number of baths, n (%) | |
| At least once daily | 328 (69.6) |
| Less than once daily | 143 (30.4) |
| Sharing bed with other children, n (%) | |
| No | 396 (84.1) |
| Yes | 75 (15.9) |
| Share bedsheets, clothes or toilet items, n (%) | |
| No | 346 (73.5) |
| Yes | 125 (26.5) |
| Fingernails always cut short, n (%) | |
| No | 442 (93.8) |
| Yes | 29 (6.2) |
| Iron clothes and bedsheets*, n (%) | |
| No | 288 (61.1) |
| Yes | 183 (38.8) |
| Pruritus in the close contacts**, n (%) | |
| No | 214 (45.4) |
| Yes | 257 (54.6) |

*Defined as at least once a month

**Defined as individuals with whom they had frequent interactions, including sharing a bed for sleeping, exchanging personal belongings, and engaging in academic and recreational activities together

## Prevalence of scabies

Overall prevalence of scabies among the orphaned children was 31.6%. According to the IACS criteria, 41.6% of them were suffering from clinical scabies while the rest of 58.4% were suffering from suspected scabies. Severity of scabies was mild, moderate and severe in 60.4%, 28.2% and 11.4% children, respectively (Table 2).

Genital and inguinal area was the most commonly reported site of scabies lesion found in almost 65% of the infected children followed by interdigital space, hands, legs and abdomen (Fig 1).

## Factors associated with scabies

We found that scabies was more prevalent in male orphans compared to their female counterparts (39.5% vs 29.3%). Moreover, children living in crowded orphanages, those who slept on floor, shared bed with other children for sleeping, shared personal items and had history of pruritus in close contacts or peers had a higher prevalence of scabies (Table 3).

In logistic regression model, it was found that male children had almost 2.76 times higher odds of being infected by scabies compared to female children (aOR 2.76, 95% CI 1.14, 7.45, p-value 0.032). Other significant risk factors of scabies were increased number of children per room (aOR 1.07, 95% CI 1.03, 1.12, p-value 0.001) and history of pruritus in the close contacts or peers (aOR 1.89, 95% CI 1.04, 3.46, p-value 0.038) (Table 3).

## Discussion

The objective of this study was to determine the prevalence and factors associated with scabies in orphaned children who are one of the major vulnerable groups for this disease. According to our findings, 31.6% orphaned children were affected by scabies, of which 60% were mild, 28% were moderate and 12% were severe. Among these children, 41.6% were clinically diagnosed and 58.4% were suspected for scabies according to the IACS criteria.

Scabies is a disease of poverty affecting predominantly children from low socioeconomic condition [22]. The orphanages might be the vulnerable sites for scabies endemic due to overcrowding and poor living circumstances. However, there is scarcity of evidence regarding the prevalence of this disease among orphaned children in Bangladesh. A recent study conducted by the senior author of this paper and others among children from Islamic residential religious schools (madrasahs) reported that almost one-third of the children were infected by scabies in these institutions, comparable to our findings [23].

Orphanages are found to be endemic places for scabies worldwide, especially in developing countries. For instance, a study from India reported scabies as one of the most common skin infections among the children and adolescents living in the orphanages [24]. Similar findings were reported from orphanages of other developing countries like Sudan and Ethiopia where a significant number of children living in these institutions were affected by scabies [25,26].

The prevalence of scabies among children found in our study, coincides with the findings from other developing countries. In neighboring India, the prevalence of scabies among school children ranged from 20 to 39% in different studies

**Table 2. Scabies related characteristics of the children living in orphanages.**

| Characteristic | n (%) |
|---|---|
| Overall prevalence (n = 471) | 149 (31.6) |
| IACS criteria (n = 149) | |
| Clinical scabies (B1/B2/B3) | 62 (41.6) |
| Suspected scabies (C1/C2) | 87 (58.4) |
| Severity of scabies (n = 149) | |
| Mild | 90 (60.4) |
| Moderate | 42 (28.2) |
| Severe | 17 (11.4) |

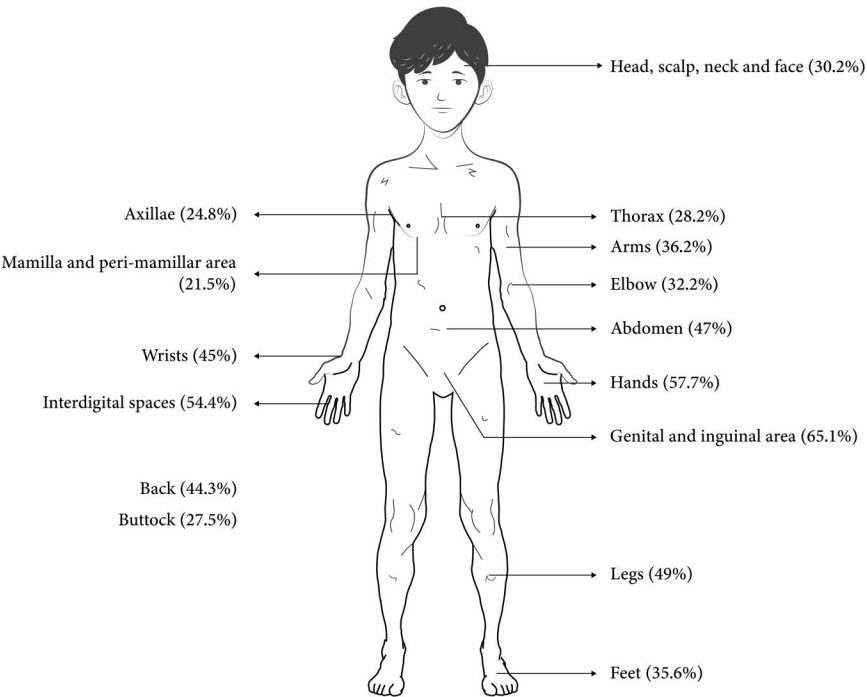

**Fig 1. Sites of lesion of scabies among the children living in orphanages (n = 149) (The clip-art used in this figure was sourced from Wikimedia Commons).**

[27,28]. In Nepal, another country from South Asian region, the prevalence was reported as 32%, similar to our finding [29]. Few countries like Pakistan reported a higher prevalence of scabies among children (almost 57%) [30]. By contrast, majority of the other lower- and middle-income countries reported a lower prevalence of scabies like Ethiopia (19%) [31], Cameroon (18%) [32], Nigeria (13%) [33] etc. However, most of these studies were conducted in community or school settings, whereas our study was conducted in an institutional orphanage setting. Children in orphanages often come from low socioeconomic backgrounds and live in crowded conditions, which facilitate close contact with infected peers and increase the risk of scabies transmission. Furthermore, as scabies was primarily diagnosed clinically in the majority of the studies, subjective variations in the diagnostic accuracy of data collectors might contribute to differences observed between the present study and previous research.

Among our participants, male children were more vulnerable to scabies infestation compared to their counterparts, a pattern that is likely a universal phenomenon [11,31,32]. These male children usually spend most of their daytime at the field playing through touching each other and handling contaminated articles with the scabies mite from their peers, which might make them vulnerable to scabies. Personal behavioral factors, such as bed-sharing, clothing-sharing, infrequent bathing, lack of soap use, etc. have been suggested as risk factors for scabies in some studies [34]. However, our logistic regression analysis did not find a significant association between these practices and scabies, which is in line with the balance of the wider evidence base. Rather than individual behaviors, the link between poverty and scabies is more likely driven by factors such as housing density and limited access to healthcare [22]. Notably, in our study, the strongest positive association with scabies was the number of children per room. As a contagious disease, the spread of scabies is exacerbated among closely resident and interacting peer groups of children, as evidenced in our study. This aspect highlights the importance of understanding not only individual-level factors but also the dynamics of social interactions in the transmission of scabies.

**Table 3. Characteristics of children with or without scabies.**

| Characteristic | Scabies | | aOR (95% CI) | p-value |
|---|---|---|---|---|
| | Yes, n=149 | No, n=322 | | |
| Age (years), mean (SD) | 11.64 (3.0) | 12.01 (3.3) | 1.01 (0.94, 1.08) | 0.801 |
| Gender, n(%) | | | | |
| Male | 43 (39.4) | 66 (60.5) | 2.76 (1.14, 7.45) | 0.032 |
| Female | 106 (29.3) | 256 (70.7) | 1.00 | |
| Number of orphans in the orphanage, median (IQR) | 146 (85.0) | 70 (84.5) | 1.01 (0.98, 1.02) | 0.856 |
| Number of children sharing same toilet, median (IQR) | 15 (13.0) | 15 (9.0) | 0.99 (0.93, 1.05) | 0.736 |
| Number of children per room, median (IQR) | 22 (15.0) | 10 (15.0) | 1.07 (1.03, 1.12) | 0.001 |
| Sleeping place, n(%) | | | | |
| On bed | 46 (22.9) | 155 (77.1) | 1.00 | |
| On floor | 103 (38.1) | 167 (61.8) | 2.07 (0.86, 5.29) | 0.112 |
| Number of baths, n(%) | | | | |
| At least once daily | 90 (27.4) | 238 (72.5) | 1.00 | |
| Less than once daily | 59 (41.3) | 84 (58.7) | 1.10 (0.61, 2.00) | 0.656 |
| Sharing bed with other children, n(%) | | | | |
| No | 140 (35.3) | 256 (64.6) | 1.00 | |
| Yes | 9 (12.0) | 66 (88.0) | 1.35 (0.45, 4.09) | 0.590 |
| Share bedding, clothes or toilet items, n(%) | | | | |
| No | 128 (36.9) | 218 (63.0) | 1.00 | |
| Yes | 21 (16.8) | 104 (83.2) | 1.23 (0.53, 2.82) | 0.603 |
| Fingernails always cut short, n(%) | | | | |
| No | 146 (33.0) | 296 (66.9) | 1.11 (0.29, 5.59) | 0.913 |
| Yes | 3 (10.3) | 26 (89.7) | 1.00 | |
| Iron clothes and bedsheets, n(%) | | | | |
| No | 119 (41.3) | 169 (58.7) | 2.03 (0.98, 4.29) | 0.059 |
| Yes | 30 (16.4) | 153 (83.6) | 1.00 | |
| Pruritus in the close contacts, n(%) | | | | |
| Yes | 115 (44.7) | 142 (55.2) | 1.89 (1.04, 3.46) | 0.038 |
| No | 34 (15.9) | 180 (84.1) | 1.00 | |

Given the high prevalence of scabies reported in our study (31.6%), relying solely on individual case management is unlikely to achieve effective disease elimination in these settings, particularly where close contact with infected individuals is frequent. Therefore, a more effective strategy such as mass drug administration (MDA) using oral ivermectin for all residents and caregivers might be considered in high-burden settings [35].

## Strengths and limitations

Despite being one of the major tropical diseases contributing to deteriorated quality of life, scabies remained a neglected area in both research and the formulation of preventive policies. In Bangladesh, a striking disparity exists as scabies lacks a comprehensive national prevention strategy at the community level, a contrast to other prevalent tropical diseases such as tuberculosis, malaria, and dengue. This study endeavors to establish a baseline understanding of scabies prevalence, focusing on a potentially high-burden context—the living conditions within orphanages. However, the study is not without its limitations. Firstly, the examination was confined to orphanages in Dhaka city, raising questions about

its generalizability to the broader national context. Secondly, the reliance on clinical diagnostic criteria, as per the IACS Criteria, introduces an element of subjectivity, dependent on the clinical acumen and expertise of the examining physician. Lastly, the examination specifically targeted limited body extremities, a methodological choice that may impact the accuracy of diagnosis and the categorization of scabies severity. These limitations warrant consideration in the interpretation of findings and underscore the need for future research endeavors to address broader demographic contexts and refine diagnostic methodologies.

## Conclusions

In conclusion, our study reported a substantial level of prevalence of scabies among orphanage children in Dhaka, Bangladesh. Major factors contributing to this prevalence include male gender, overcrowded living conditions, and close contact with infected peers. Addressing this public health concern necessitates a comprehensive approach that includes interventions aimed at reducing the number of children per room and strategically minimizing contact with individuals affected by scabies when they are infective. Additionally, considering the high prevalence of scabies in the orphanages, a mass treatment program might better control the transmission of the disease. Finally, the findings of our study would guide the implementation of targeted strategies to mitigate scabies transmission within these settings, thus fostering a healthier environment for the vulnerable population of children in orphanages.

## Supporting information

**S1 File. Questionnaire (English).**
(DOCX)

**S1 Data. Dataset.**
(XLSX)

## Acknowledgments

We acknowledge contribution of Dr. Taha Choudhury, Dr. Susmita Zaman, Dr Salwa Islam, Dr Paromita Zaman for their contribution in data collection process. The authors would like to express their sincere gratitude to Pi Research & Development Center, Dhaka, Bangladesh (www.pirdc.org), for their help in manuscript revision and editing.

## Author contributions

**Conceptualization:** Md Abdur Rafi, Mohammad Jahid Hasan.

**Data curation:** Md Abdur Rafi.

**Formal analysis:** Md Abdur Rafi.

**Funding acquisition:** Md. Golam Hossain.

**Investigation:** Md. Rashidul Hasan, Mohammad Jahid Hasan.

**Methodology:** Md Abdur Rafi, Mohammad Jahid Hasan.

**Project administration:** Mohammad Jahid Hasan.

**Software:** Md Abdur Rafi.

**Supervision:** Md. Golam Hossain, Mohammad Jahid Hasan.

**Validation:** Md. Golam Hossain.

**Visualization:** Md Abdur Rafi.

**Writing – original draft:** Md Abdur Rafi, Mohammad Jahid Hasan.

**Writing – review & editing:** Md Abdur Rafi, Md. Golam Hossain, Md. Rashidul Hasan, Mohammad Jahid Hasan.

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
