## [Decision Letter · Decision Letter 0]

17 Mar 2025

Prevalence and risk factors of scabies among orphans: a cross-sectional study in Bangladesh

Dear Dr. Hasan,

Thank you for submitting your manuscript to PLOS Neglected Tropical Diseases. After careful consideration, we feel that it has merit but does not fully meet PLOS Neglected Tropical Diseases's publication criteria as it currently stands. Therefore, we invite you to submit a revised version of the manuscript that addresses the points raised during the review process.

Please submit your revised manuscript within 60 days May 16 2025 11:59PM. If you will need more time than this to complete your revisions, please reply to this message or contact the journal office at plosntds@plos.org. Please include the following items when submitting your revised manuscript:

We look forward to receiving your revised manuscript.

Kind regards,

Jo Middleton (NIHR Global Health Research Unit on NTDs) 

Guest Editor

Justin Remais

Section Editor

Shaden Kamhawi

co-Editor-in-Chief

Paul Brindley

co-Editor-in-Chief

**Additional Editor Comments :**

We have now, after some delay, received all reviewer comments. Whilst they are generally positive, as you can see from reading them our reviewers still considered your manuscript needs further work. In addition to the comments provided below, I would also like to flag up that whilst the manuscript is generally well written, its grammar and word use could be improved. One reviewer specifically asked me to suggest to you that you should pay further attention to this, so as to improve the manuscript for the reader.

Note, one of the reviewers (reviewer 1) is a medical statistician (with a background in scabies work), please follow their guidance in your revision of the manuscript, and if you think it contradicts anything said by one of the other reviewers please follow reviewers 1’s instructions which take primacy.

In addition to the reviewer comments, there are a few issues which I as editor would like you to address. I outline these below.

**RE: SARCOPTES SCABIEI VAR. HOMINIS**

Line 44: ‘human parasite mite Sarcoptes scabiei var. hominis.’

Change text to: ‘parasitic mite *Sarcoptes scabiei* ’. The var. hominis and reference to ‘human’ is outdated and should be removed. We have known for a decade that the phylogenetics of S. scabiei do not support either (1) a uniform host-specific set of variants (as indicated in your manuscript) or (2) panmixia, in which all S. scabiei clades – across host species – interbreed. Make this change, and if interested, see the following paper (of which I am not an author or otherwise involved): Andriantsoanirina V, Ariey F, Izri A, Bernigaud C, Fang F, Charrel R, Foulet F, Botterel F, Guillot J, Chosidow O, Durand R. Sarcoptes scabiei mites in humans are distributed into three genetically distinct clades. Clin Microbiol Infect. 2015 Dec;21(12):1107-14. doi: 10.1016/j.cmi.2015.08.002.

**RE: PARTIAL EXAMINATIONS AND REPORTING OF SCABIES SIGN BODY DISTRIBUTION**

In pages 115-116 you write: “examination of sensitive areas such as the groin, buttocks, breasts and torso were skipped as it was not practical in the field setting.”

Please provide a justification for why this was not considered practical in a field setting. Many epidemiological studies on scabies, including ones I've been involved with personally in both low and high income settings, have carried out this in the field. Notably you use IACS criteria for clinical scabies which requires at least one of the following three to be present: “B1: scabies burrows; B2: Typical lesions affecting male genitalia; B3: Typical lesions in a typical distribution and two history features.” If you didn't examine male genitalia, how can you say lesions there are typical or not (and as I outline in a minute, you do report that many of your study participants had scabies signs there). In lines 153-154 you write: ‘Genital and inguinal area was the most commonly reported site of scabies lesion found in almost 65% of the infected children followed by interdigital space, hands, legs and abdomen’. This is also shown in figure 1. Given you state that you didn't in fact examine genitalia, but genitalia was the most commonly reported site, explain this contradiction. Was this based on patient/other self report? If so do your records disaggregate distribution of scabies signs between confirmed signs by examination and declared signs by patient/other report? If it does, please rectify (and disaggregate) your text and Figure 1 so this is clear to the reader. If it doesn't, provide justification for how your reported scabies sign distribution statistics can be reliably interpreted.

**RE: DATA COLLECTION MATERIALS**

Include in supplementary material an English translation of the questionnaire, and any other data collection forms used (for examination etc.), so readers can know what was actually asked.

**RE HYGIENE**

Throughout the manuscript there are a number of mentions of personal hygiene. In most cases you do not define exactly what you mean and imply a relationship to scabies prevalence or risk which was not found in your own study (despite looking for it), and is not supported by the existing balance of the overall evidence base or the International Alliance for the Control of Scabies. For example, on line 212-215 you write: “Personal hygiene is one of the major influencing factors of scabies among children. It is evidenced that children who maintain less personal hygiene, and use shared personal stuffs have a higher odd of having scabies [29]”. Similarly, on line 224-226 you state: “Though the significance of personal hygiene in scabies incidence is widely acknowledged [29], our study did not distinctly establish a conclusive association with hygiene practices.” In fact, the reference you give to support these two statements ( [29] Engelman et al. 2019. Lancet) says the opposite of what you claim re personal hygiene. Its only discussion of hygiene is as follows: “Scabies is often incorrectly attributed to poor hygiene, which can lead to stigma, shame, and reduced health seeking. However, hygiene and handwashing do not affect the mite or transmission,95 and highly effective control has been shown without any measures addressing hygiene or environment (see section on population-level control),96 suggesting that associations with situations of poverty and disadvantage are probably due to poor access to health care and treatment, or the effects of overcrowding.21,97”

*****Please make the following changes in your updated manuscript, or provide a strong point-by-point defence for why not.*****

Line 59: Remove “hygiene conditions”. No reference is provided for this statement .

Line 59-61: You write: “Overcrowded living conditions, shared sleeping arrangements, sharing of clothes and towels, poor hygiene, malnutrition, and travel to scabies outbreak areas are common risk factors for scabies [9, 10].” Remove “poor hygiene” from this list. The results of the study you give as reference 9 (Melese et al. 2023. BMC Trop Med Health) does support your inclusion of clothes sharing (at least re sharing with a scabies case), hence I am not saying to remove that, but they do not report results showing ‘personal hygiene’ is a risk factor. Your reference10 (Azene et al 2020. BMC Infect Dis) does not support your statement, in fact the opposite. Its results state: (A) “Association between frequency of bath and scabies infestation… the random effect model of a pooled odds ratio of frequency of the bath were not statically significant (OR: 1.47, 95% CI: 0.59, 3.64)”. (B) “Association of washing hand with soap and scabies infestation… The overall pooled odds ratio result of this study revealed that washing hands with soap had no statistically significant association with scabies infestation (OR: 0.89, 95%CI: 0.57, 1.40)”.

Line 69: Remove “limited access to proper hygiene facilities”. Whilst your reference 6 [Cox et al 2020. B J Dermatol) does discuss communal settings, so is appropriate for this section, it does not mention “limited access to proper hygiene”.

Line 184: “A recent study conducted among children from islamic residential religious schools reported that almost one third of the children were infected by scabies in these institutions, comparable to our findings [19]. The difference in living condition and hygiene practice among children from different institutions might cause the difference.” Firstly, if you are going to mention this study, for transparency I would like you to insert as follows (capitalisation just to emphasise my addition): “A recent study conducted BY THE SENIOR AUTHOR OF THIS PAPER AND OTHERS among children…” Secondly, remove “and hygiene practice”. As outlined above, this is not supported by the overall evidence base, and infact the study reported in your reference 19 (by one of your own authors) also did not to find any statistically significant association despite between personal washing and scabies, despite looking for it. For example, in Table 3 of Reference 19 its states: “Irregular bathing (yes vs noR) p=0.555; Use soap for baths (yes vs noR) p=0.260” etc.

Line 198-201: “However, majority of these studies were conducted in community or school setting, while our study was conducted in an institutional setting of orphanages where children belonged to a low socioeconomic condition with crowded living and deprived of adequate personal hygiene and healthcare access.” Given what is outlined above, remove “personal hygiene”.

Line 212: “Personal hygiene is one of the major influencing factors of scabies among children. It is evidenced that children who maintain less personal hygiene, and use shared personal stuffs have a higher odd of having scabies [29], though in the present study, we were not able to demonstrate such association of scabies with factors regarding personal hygiene.” I outline above why your reference 29 says the opposite of what you claim it does, so make the following changes so it accurately reflects reference 29, and your own results. Suggested edit (capitalisation just to emphasise my additions): “Personal BODY WASHING is OFTEN PRESUMED BY LAYPEOPLE AND SOME HEALTH WORKERS TO BE A major influencing factor of scabies. HOWEVER, OUR STUDY’S LOGISTIC REGRESSION MODEL FOUND NO ASSOCIATION BETWEEN SCABIES AND BATHING, WHICH IS IN LINE WITH THE BALANCE OF THE WIDER EVIDENCE BASE.[29] INSTEAD OF PERSONAL WASHING, CONNECTIONS BETWEEN POVERTY AND SCABIES LIKELY STEM FROM HOUSING DENSITY AND LIMITED ACCESS TO HEALTHCARE.[29] NOTABLY, IN OUR STUDY THE STRONGEST POSITIVE ASSOCIATION WITH SCABIES WAS NUMBER OF CHILDREN PER ROOM.”

Lines 250-256 (from your conclusion): “Major factors contributing to this prevalence include male gender, overcrowded living conditions, and close contact with infected peers.” The first sentence here is good, and reflects your findings. However, most of the second sentence does not flow from your findings at all (which should be the basis of your conclusion). You write: “Addressing this public health concern necessitates a comprehensive approach involving interventions aimed at enhancing the hygienic behaviors of children, promoting health education initiatives, and strategically minimizing contact with individuals affected by scabies.” Remove reference to hygiene (looked for, but not demonstrated by your study) and health education (you do not report any survey of knowledge of scabies, so should not recommend education as a panacea). Replace with: “Addressing this public health concern necessitates a comprehensive approach involving interventions aimed at REDUCING DENSITY OF CHILDREN PER ROOM, and strategically minimizing contact with individuals affected by scabies WHEN THEY ARE INFECTIVE.” In addition, I would suggest (though do not require) that you add in something emphasizing the need for mass treatments in affected orphanages, as the prevalence you report (31.63%) is too high to eradicate successfully though individual case management (particularly given that patients will be infective for a substantial part of the initial asymptomatic incubation period). For more information on this see: Engelman et al. 2021. A framework for scabies control. PLoS Negl Trop Dis. doi: 10.1371/journal.pntd.0009661. (NOTE: I am not an author of that paper, so am not recommending something by myself!). The final sentence starting “The findings” is fine as is.

**RE DUPLICATION OF CONTENTS**

Two paragraphs in your discussion section seem to be mainly two different drafts of the same paragraph, covering the same topics and using the same references at the same points, but worded slightly differently. I suspect this is just a simple copy-paste editing mistake. Retain the paragraph on lines 204-215 (making the changes outlined above re hygiene etc.), and delete the following lines in the subsequent paragraph: 216-225.

Then, on line 226 delete “association with hygiene practices. Moreover, being”, and instead start the sentence as follows (capitalisation to indicate my insertions only): “AS a contagious disease, THE spread of scabies is exacerbated among CLOSELY RESIDENT AND INTERACTING peer groups of children, AS evidenced in our study. This aspect highlights the importance of understanding not only individual-level factors but also the dynamics of social interactions in the transmission of scabies.”

****I personally feel your study is a useful addition to knowledge of a medically neglected problem affecting the highly disadvantaged; hence my support for your paper but also my attention to its details when you misrepresent that which you are citing, or make recommendations not based on your study findings. I look forward to reading your revised manuscript.****

**Journal Requirements:**

1) Please provide an Author Summary. This should appear in your manuscript between the Abstract (if applicable) and the Introduction, and should be 150-200 words long. The aim should be to make your findings accessible to a wide audience that includes both scientists and non-scientists. Sample summaries can be found on our website under Submission Guidelines:

Potential Copyright Issues:

i) Figure 1. Please confirm whether you drew the images / clip-art within the figure panels by hand. If you did not draw the images, please provide (a) a link to the source of the images or icons and their license / terms of use; or (b) written permission from the copyright holder to publish the images or icons under our CC BY 4.0 license. Alternatively, you may replace the images with open source alternatives. See these open source resources you may use to replace images / clip-art:

5) In the online submission form, you indicated that "Patient-level data will be available on request from the corresponding author." All PLOS journals now require all data underlying the findings described in their manuscript to be freely available to other researchers, either

1. In a public repository

2. Within the manuscript itself

3. Uploaded as supplementary information.

2) State what role the funders took in the study. If the funders had no role in your study, please state: "The funders had no role in study design, data collection and analysis, decision to publish, or preparation of the manuscript.".

**Reviewers' Comments:**

Reviewer's Responses to Questions

**Key Review Criteria Required for Acceptance?**

**Methods**

-Are the objectives of the study clearly articulated with a clear testable hypothesis stated?

-Is the study design appropriate to address the stated objectives?

-Is the population clearly described and appropriate for the hypothesis being tested?

-Is the sample size sufficient to ensure adequate power to address the hypothesis being tested?

-Were correct statistical analysis used to support conclusions?

-Are there concerns about ethical or regulatory requirements being met?

Reviewer #1: The sample size was based on estimating the proportion of children with scabies with a gievn precision, which is appropriate. It should be noted that the modelling analyses are exploratory, as the study samples size was not based on the modellling.

Univariable testing should be removed as it is redundant thanks to the regression modelling (see results comments).

There is no reason to arbitrarily set a threshold for "statistical significance". This phrase should not be used, and p-values should be interpreted as strength of evidence against the relevant null hypothesis, rather than dichotomised. This doesn't seem to have been done at all in the paper anyway, so "A value of<0.05 using a 95% confidence interval in logistic regression model was defined as statistically significant level." can be removed from the methods.

Need details on data collection - paper CRFs -> databse? What database software?

Reviewer #2: The objectives of the study are clearly articulated with a clear testable hypothesis stated. The study design is appropriate to address the stated objectives and the population is clearly described and appropriate for the hypothesis being tested.

The sample size was sufficient to ensure adequate power to address the hypothesis being tested and correct statistical analysis was used to support conclusions. There are no concerns regarding ethical or regulatory requirements.

Reviewer #3: Yes. The study design is suitable for describing the prevalence and the risk factors of scabies in orphanages in the Dhaka city of Bangladesh. It included 471 children from 5 orphanages, which was determined through a sample size estimation. They used the latest criteria of the International Alliance for the Control Scabies, but they made it feasible for its use on the field by only selecting the clinical and suspected scabies criteria. This also applies for the reasoning of not examining some sensible body parts, which then was explained and supported with evidence.

**Results**

-Does the analysis presented match the analysis plan?

-Are the results clearly and completely presented?

-Are the figures (Tables, Images) of sufficient quality for clarity?

Reviewer #1: Results

Baseline characteristics and Table 1

Mean and SD have been used to summarise all continuous variables, but it’s unclear if these are appropriate, as a number of the variables are counts. Counts are often not normally distributed, so median and interquartile range may be more appropriate.

I think there would be value to showing distribution graphs for these characteristics, so the reader can see the distributions for themselves.

The second heading for table 1 is “n (%)”, but many of the quantities given are mean (SD) – the heading should be changed to reflect this. Add % to any percentage, this will make it obvious when summaries are proportions. Where means or percentages are given with decimal places, just show 1, 2 is unnecessary (same in other tables, but give p-values to 3 d.p.).

Factors associated with scabies

The t-tests and chi-square tests are redundant as the results from the regression model are superior. It’s also unclear if the assumptions are met for all of these tests (particularly the normality assumption for t-tests and minimum of 5 expected events (scabies cases) per cell for chi-square). P-values should be removed from Table 3, just summaries and 95% CIs are sufficient.

Logistic regression model

11 variables are included in the model with 149 cases of scabies, which is appropriate. Has collinearity of these variables been checked though? Several of these variables sound like they could be colinear. To check this in Stata you can fit a linear regression model with scabies as the outcome and all the predictor variables, followed by the ‘vif’ command (it does not matter that the outcome is binary, this only considers the predictors). You want to see VIFs <10, otherwise you’ll need to play around with the combination of predictors to identify which are causing high VIFs.

I also think that site (orphanage) should be included in the model to adjust for differences between sites.

Reviewer #2: The analysis presented matched the analysis plan and the results are clearly and completely presented. The figures (Tables, Images) have sufficient quality for clarity.

Reviewer #3: Yes, the results are clearly and completely presented. They allowed the identification of the prevalence and of the risk factors of scabies in the orphanages in the Dhaka city of Bangladesh. The prevalence was similar as the one reported in other studies, including one that was made in the same city, but in another environment: https://pubmed.ncbi.nlm.nih.gov/38942588/ The identification of the risk factors allows to develop tailored interventions, which is crucial to tackle its burden. The manuscript only has 1 figure and several tables, but they serve its purpose.

**Conclusions**

-Are the conclusions supported by the data presented?

-Are the limitations of analysis clearly described?

-Do the authors discuss how these data can be helpful to advance our understanding of the topic under study?

-Is public health relevance addressed?

Reviewer #1: The conclusions seem generally appropriate.

Some of the predictors do need explicit definitions though -

Iron clothes and bedsheets - how often?

Pruritus in the close entourage - what defines "close entourage"?

Reviewer #2: The conclusions are supported by the data presented. The limitations of analysis and study are clearly described.

The authors discuss how these data can be helpful to advance our understanding of the topic under study and the public health relevance.

Reviewer #3: The conclusions are supported by the data presented and they propose interventions to stop the chain of transmission. Orphanages are environments in which children are greatly exposed to different diseases. It is important to build specific strategies to approach and manage each of them. This study does that. The authors understood the problem and they conducted it in an operative way. They clearly describe its limitations and they do not pretend to make them generalizable for other cities and countries. Local research is valuable. It has public health implications.

**Editorial and Data Presentation Modifications?**

Reviewer #1: The English in the paper is generally good, but there are some odd phrases in some places, such as “personal staffs”, “toilet stuffs”.

“sharing room with a greater number of children” – unclear what this means

“Fingernail always cut short” -> “Fingernails always cut short”

Reviewer #2: This is a good manuscript with interesting and relevant findings of a neglected tropical disease, affecting most people in low socioeconomic settings of the world.

There is need to make some corrections/amendments to the manuscripts as suggested below:

Line 22: consider changing 'condition' to "setting or status or background". The follwoing sentence change 'The objective of the present study was to' to "This study was conducted to".

Line 25: consider moving parts of the first sentence to results section, i.e. '471 children' and 'five orphanages'. Replace the sentence with "This was a cross-sectional study among children in selected orphanages in Dhaka city of Bangladesh."

Line 30: consider starting the section with this sentence: "471 children living in five selected orphanages of Dhaka city of Bangladesh were screened for scabies. Majority..."

Line 36: consider removing 'Almost'

Line 44: consider adding reference to this sentence; you may wish to move the reference in Line 45 to here

Line 62: consider replacing 'brackets' with "settings"

Line 86: consider replacing 'The present one' with "This"; remove 'five' from this sentence - it should be in the results section.

Lines 100-103: consider moving these lines to results section

Line 139: consider replacing 'the present' with "this"

Lines 143 and 161: consider replacing 'staffs' with "stuff"

Line 162: consider replacing 'entourage' with "company"

Lines 163-170: consider correcting the number on the p-value to 2 decimal places except in very small number (significant result ~ <0.001, <0.003)

Line 173: consider replacing 'the present' with "this"

Line 175: consider adding "s" to 'finding'; remove 'almost'; add "of which" after 'affected by scabies,'

Lines 178-181, 232-247: consider adding references to these sentences

Reviewer #3: The manuscript is fluent and well-written and it does only require some minor corrections:

Line 2. "bangladesh" should be written with an upper case letter: "Bangladesh".

- Line 146. "Scabies" should be written in lower case letters: "scabies".

- Line 241. The "International Alliance for the Control of Scabies (IACS) Criteria" has already been mentioned. It should say "IACS Criteria".

**Summary and General Comments**

Reviewer #1: I have primarily reviewed the statistics - this paper provides a good summary of the population of interest and answers the question it set out to - the prevalence of scabies in this population. The exploratory logistic regression model is credible (but does require examination of collinearity and inclusion of site) and the results are interpreted appropriately.

Reviewer #2: This manuscript describes an important study on a highly prevalent NTD which affects the most vulnerable and low socioeconomic people in remote, limited resource settings. This study has interesting and significant findings which are very relevant to Health authorities and policy makers in most endemic areas facing the same and similar disease (s).

Reviewer #3: I found this study relevant for public health. Addressing public health problems with general health recommendations that have been made elsewhere is not desirable nor suitable. They can be used to build a framework, but local research is required to tailored them. This manuscript is the perfect example of how to do that. I do hope that the study is replicated in other parts of the country and that it help to improve the health of children who live in orphanages. This neglected tropical disease can be controlled.

PLOS authors have the option to publish the peer review history of their article (what does this mean? ). If published, this will include your full peer review and any attached files.

**Do you want your identity to be public for this peer review?** For information about this choice, including consent withdrawal, please see our Privacy Policy .

Reviewer #1: **Yes: ** Christopher Jones

Reviewer #2: No

Reviewer #3: No

**Figure resubmission:**

**Reproducibility:**



---

## [Decision Letter · Decision Letter 1]

21 Oct 2025

Dear Dr Hasan,

We are pleased to inform you that your manuscript 'Prevalence and risk factors of scabies among orphans: a cross-sectional study in Bangladesh' has been provisionally accepted for publication in PLOS Neglected Tropical Diseases.

Best regards,

Shaden Kamhawi

co-Editor-in-Chief

Paul Brindley

co-Editor-in-Chief

Reviewer's Responses to Questions

**Key Review Criteria Required for Acceptance?**

**Methods**

-Are the objectives of the study clearly articulated with a clear testable hypothesis stated?

-Is the study design appropriate to address the stated objectives?

-Is the population clearly described and appropriate for the hypothesis being tested?

-Is the sample size sufficient to ensure adequate power to address the hypothesis being tested?

-Were correct statistical analysis used to support conclusions?

-Are there concerns about ethical or regulatory requirements being met?

Reviewer #1: Following revisions, I think the key criteria for acceptance are met.

Reviewer #3: The authors did implemented the modifications that were requested in this section, specially those that were requested by reviewer 1. The supplementary file 1 has been written in English. They provided a detailed description of the methodology that they used for performing the examination of sensitive areas. This was one of the main points of confusion in the previous version, but I consider that it has been adequately explained in this one. They rewrote the methodology of the statistical analysis.

**Results**

-Does the analysis presented match the analysis plan?

-Are the results clearly and completely presented?

-Are the figures (Tables, Images) of sufficient quality for clarity?

Reviewer #1: Results are presented appropriately.

Reviewer #3: The format of presentation was standardized. The authors included median and interquartile ranges to better describe the distribution of the results. They justified the reasons of keeping some variables in the final model. They are presented in table 3. They did left the p-values, but provided the reasoning behind that.

**Conclusions**

-Are the conclusions supported by the data presented?

-Are the limitations of analysis clearly described?

-Do the authors discuss how these data can be helpful to advance our understanding of the topic under study?

-Is public health relevance addressed?

Reviewer #1: Conclusions are appropriate.

Reviewer #3: The conclusions had major modifications. Those were made to fulfill the requirements of reviewer 1. The description of the results has improved with them. The authors maintained the description of the personal behavioral factors, but they explained that this was required to accurately reflect the current evidence-based recommendations. They eliminated the hygiene and health education reference to stick with what was presented in their study. They eliminated the deduplicated paragraphs that described a similar idea.

**Editorial and Data Presentation Modifications?**

Reviewer #1: (No Response)

Reviewer #3: All the modifications that were requested were implemented. I recommend its publication in its current form.

**Summary and General Comments**

Reviewer #1: (No Response)

Reviewer #3: This improved version of the manuscript is valuable for public health as it provides an insight of the transmission of the disease in a particular setting, in a specific country. The results are not generalizable, but they can guide local responses and other researchers may replicate it in their specific conditions. It was adequately-designed and appropriately-conducted. This is the kind of research that can shape public health policies when replicated and escalated.

PLOS authors have the option to publish the peer review history of their article (what does this mean? ). If published, this will include your full peer review and any attached files.

**Do you want your identity to be public for this peer review?** For information about this choice, including consent withdrawal, please see our Privacy Policy .

Reviewer #1: **Yes: ** Christopher Jones

Reviewer #3: No

---

## [Editor Report · Acceptance letter]

Dear Dr Hasan,

We are delighted to inform you that your manuscript, "Prevalence and risk factors of scabies among orphans: a cross-sectional study in Bangladesh," has been formally accepted for publication in PLOS Neglected Tropical Diseases.

Best regards,

Shaden Kamhawi

co-Editor-in-Chief

Paul Brindley

co-Editor-in-Chief
